# Behavioral and Physiological Differences in Female Rabbits at Different Stages of the Estrous Cycle

**DOI:** 10.3390/ani13213414

**Published:** 2023-11-03

**Authors:** Xin Chen, Rongshuai Jin, Anqi Yang, Jiacheng Li, Ying Song, Bohao Zhao, Yang Chen, Xinsheng Wu

**Affiliations:** College of Animal Science and Technology, Yangzhou University, Yangzhou 225009, China; 201001105@stu.yzu.edu.cn (X.C.); rsjinnn@163.com (R.J.); 201902123@stu.yzu.edu.cn (A.Y.); 201902109@stu.yzu.edu.cn (J.L.); 201902317@stu.yzu.edu.cn (Y.S.); bhzhao@yzu.edu.cn (B.Z.)

**Keywords:** female rabbits, estrus, behavior, hormone, follicle

## Abstract

**Simple Summary:**

Estrus involves complex physiological signs and behavioral changes, but the exact ovulation time is difficult to determine, which is very important to improve the fertility of rabbits. This study aimed to analyze female rabbits’ changing behavior and physiology at different stages of the estrous cycle. The female rabbits showed less foraging, drinking, grooming, and biting while in estrus. Related physiological hormones (FSH, LH, P_4_, and E_2_) and follicular development at different estrous stages also differed. The number of primary follicles in estrus was significantly higher than in the other stages. Some genes, such as *CYP19A1* and *IGF1R*, related to the estrous cycle in female rabbits were screened via ovarian transcriptome analysis. Overall, extensive information relating the behaviors, hormones, and follicle development in female rabbits at different estrus stages was investigated for appropriate postpartum re-estrus management strategies.

**Abstract:**

Estrus involves a series of complex physiological signs and changes in behavior before ovulation, which play a crucial role in animal reproduction. However, there have been few studies that evaluate behaviors during the different stages of estrus cycle in female rabbits. Therefore, more detailed information is needed on distinguishing the various stages of the estrous cycle. This study explored the behavioral and physiological differences at various estrous cycle stages in female New Zealand White rabbits. The continuous recording method was employed to record the daily behaviors of twenty postpartum female rabbits during the estrous cycle. Compared with the diestrus stage, the duration of foraging and drinking behavior in estrus decreased significantly, and the frequency of grooming and biting behaviors increased (*p* < 0.05). Differences in reproductive hormone levels (FSH, LH, P_4_, and E_2_) and follicle development were measured at each stage via ELISA and HE staining. The FSH and LH levels showed an increasing trend and then decreased, with the lowest being in late estrus (*p* < 0.05). The P_4_ level was the lowest in estrus (*p* < 0.05), and E_2_ showed a gradually increasing trend. There was no significant difference in the number of primordial follicles at each stage, but the number of primary follicles in estrus was significantly higher than at the other stages (*p* < 0.05). To further understand the molecular regulation mechanism of the estrous cycle in female rabbits, we analyzed the ovarian transcription patterns of female rabbits in diestrus (D group) and estrus (E group) employing RNA-seq. A total of 967 differentially expressed genes (DEGs) were screened from the ovaries of female rabbits between the diestrus and estrus groups. A KEGG analysis of DEGs enriched in the estrogen signaling pathway, aldosterone synthesis, and secretion pathway, such as *CYP19A1* and *IGF1R*, was performed. The rabbits’ behavior, related physiological hormones, and molecular regulation also differed at different estrous cycle stages. The results provide recommendations for the adequate management practices of postpartum re-estrus and breeding female rabbits.

## 1. Introduction

Estrus, as the initial stage of animal reproduction, holds significance in animal conception, as it largely determines the probability of female rabbit pregnancy. However, various endogenous hormones and exogenous factors influence the presence of uncontrollability in reproductive efficiency. The estrous cycle of female rabbits is divided into diestrus, early estrus, estrus, and late estrus. The conception rate during the estrous cycle is the highest [1]. The estrus identification and timely mating of female rabbits are the key factors affecting the modern intensive production system efficiency.

During estrus, the female rabbit’s appetite decreases, the mood becomes agitated, and it will sniff around and dig nest burrows among the range of activities [2]. Typically, the female rabbit actively touches the male rabbit and rubs objects with her jaws to deposit odorous secretions, which is thought to be a scent-marking behavior [3,4]. However, the intensive production system compresses the living space of female rabbits and changes the internal physiological environment, leading to changes in the behaviors of female rabbits [5]. Therefore, researchers can acquire more precise behavioral information by synthesizing female rabbit behavior throughout several estrus stages, thereby establishing a basis for advancing reliable techniques for assessing estrus behaviors in female rabbits.

The estrous cycle of a female rabbit is closely related to the endocrine system. Reproductive hormones act as messengers to regulate the coordination of various systems and organs to maintain homeostasis [6]. *Follicle-stimulating hormone* (FSH) can promote the rapid development of follicles after the action of dominant follicles [7]. Subsequently, the increase in estradiol (E_2_) levels inhibits the secretion of FSH through negative feedback so that non-dominant follicles cannot enter the rapid growth stage. Progesterone (P_4_), a natural hormone secreted by the ovary’s corpus luteum, inhibits follicular development and steroidogenesis in ovaries [8]. When the dominant follicle develops, the luteinizing hormone (LH) level increases with E_2_, which is the key condition for follicle maturation [9,10,11]. The dynamic changes in hormones and follicular development in female rabbits at different estrus stages are still unknown.

The molecular regulation mechanism of the mammalian estrous cycle is very complex, involving several key genes and signaling pathways. The expression of *BMPR1B* in granulosa and theca cells is the most prominent in follicular development, which may be influenced by FSH and LH [12]. *CYP19A1* may regulate the estrous cycle by mediating reproductive hormone synthesis and follicular development through positive or negative regulation [13,14]. *CYP1A1* is expressed the most during ovulation in the ovary, indicating an increased hydroxylation of estrogen and the elimination of negative estrogen feedback control over LH secretion, which play direct or indirect roles in ovulation [15]. The key regulating factors in female rabbits at different estrus stages must be explored.

To investigate the changes in the behavior and physiology of female rabbits at different stages of the estrous cycle, we compared and studied the daily behaviors, hormones, follicle development, and differentially expressed genes (DEGs) in female rabbits at various estrus stages. The parameters regulating the estrous cycle of female rabbits were investigated using behavioral, endocrine, and molecular methods to provide the foundation for further research into their reproductive potential.

## 2. Materials and Methods

### 2.1. Animals

The present work acquired ethical approval from the Animal Protection and Use Committee of Yangzhou University (no. 202109221) and was conducted following rigorous adherence to animal management regulations. A total of 20 female New Zealand White rabbits, aged 8 months, were selected based on their similar body size and overall good health in the third parity.

Twenty postpartum female rabbits were transferred to individual cages (70 cm × 90 cm × 50 cm) with the camera shooting and allowed to rest for 1 day to adapt to the new environment. On the 2nd day after the transfer, light stimulation was used to induce estrus from 7:00 a.m. to 10:00 p.m. every day. All female rabbits’ daily behaviors (diestrus, early estrus, prosperous estrus, and late estrus) were monitored and recorded at 8:00 a.m. and 8:00 p.m. for 15 days. Then, the blood samples were collected from the female rabbits at different estrus stages at 7:00 a.m. every day during the 2nd estrous cycle. During the 3rd estrous cycle, ovarian tissue was collected at different estrus stages.

### 2.2. Estrus Identification

The estrus state of the female rabbits was determined according to the vaginal mucosa color daily. There are four estrus stages in the entire estrous cycle. As shown in Figure 1, the vaginal mucosa in diestrus appears pale white, pink in early estrus, dark red in estrus, and purple–black in late estrus. During the 15-day observation period, after judgment, diestrus lasted for 6 days, early estrus lasted for 4 days, estrus lasted for 2 days, and late estrus lasted for 3 days. All samples were collected in the middle of each stage.

### 2.3. Behavior Observation and Recording

A total of four surveillance cameras were strategically positioned at a distance of 2.6 m from the cage walls, ensuring an equal spacing of 3 m between each camera. The orientation of the cameras was modified to ensure a comprehensive coverage of the actions of the five female rabbits by each camera. The estrous cycle of the female rabbits was observed and documented bi-daily, with continuous monitoring conducted throughout the day, resulting in the storage of comprehensive records for all the female rabbits. The frequency and duration of each behavior were observed using a continuous recording method for 15 days. The software used to watch the video was PotPlayer (1.7.21796), which individuals watch at double speed and record individually. The number of behaviors occurring per unit of time was recorded as the frequency, while duration referred to the time from occurrence to the completion of a certain behavior. Before the formal recording of the observation, the female rabbits were pre-observed for 3 days. The specific behaviors and definitions of the female rabbits are listed in Table 1.

### 2.4. Plasma Collection and Reproductive Hormone Detection

Female rabbit blood was obtained from the auricular vein, centrifuged at 1500 r/min for 5 min, and the supernatant was absorbed. The contents of LH, FSH, P_4_, and E_2_ in the plasma were determined using enzyme-linked immunosorbent assay (ELISA) kits for rabbits, FSH kit (ml027868, Mlbio, Shanghai, China), LH kit (ml027918, Mlbio, Shanghai, China), P_4_ kit (ml028074, Mlbio, Shanghai, China), and E_2_ kit (ml027864, Mlbio, Shanghai, China). Each well’s absorbances (OD values) at 450 nm were measured using a microplate reader (Infinite F50, TECAN, Männedorf, Switzerland) with the blank as a zero reference. The sample concentration was calculated according to the standard curve.

### 2.5. Hematoxylin and Eosin (HE) Staining, and Follicle Counts

Ovaries were collected and fixed with 4% paraformaldehyde. The tissue size was trimmed and dehydrated with ethanol and xylene at different concentration gradients, tissue wax was immersed and embedded using an embedding machine, and blocks were sliced with a thickness of 5 μm using a microtome. The prepared sections were then dewaxed, rehydrated, stained with HE, sealed with neutral gum, observed, and photographed. The number of follicles (primordial follicle, primary follicle, secondary follicle, and tertiary follicle) was observed and counted.

### 2.6. RNA-Seq Analysis

The ovaries of female rabbits in the diestrus (D1–D3) and estrus (E1–E3) stages were collected for RNA-seq analysis. After total RNA isolation and library preparation, next-generation sequencing (NGS) based on the Illumina sequencing platform was used. The filtered reads were compared with rabbit genomes using HISAT2(2.1.0) software. Gene expression was calculated using fragments per kilobase of transcript per million mapped reads (FPKM). DESeq was used to analyze the differential expressions between groups. |log_2_FoldChange| > 1 and *p*-value < 0.05 were set as conditions of screening DEGs. Gene ontology (GO) and Kyoto Encyclopedia of Genes and Genomes (KEGG) enrichment analysis were performed for functional annotation and pathway analyses of the DEGs.

### 2.7. Real-Time Quantitative PCR (RT-qPCR)

Total RNA was extracted using TRIzol (Invitrogen, Carlsbad, CA, USA) following the manufacturer’s instructions. Real-time PCR was performed using ChamQ™ SYBR^®^ qPCR Master Mix (Vazyme, China) on a QuantStudio^®^ 5 Real-Time PCR System. The primer sequences are listed in Table 2. Each sample was examined three times, and the data were normalized to GAPDH. The relative gene expression level was calculated using the 2^−ΔΔCt^ method.

### 2.8. Statistical Analysis

The differences in behavior, reproductive hormone levels, and follicle count were analyzed utilizing a one-way ANOVA using SPSS (ver. 25.0) software. The RT-qPCR data were analyzed using a *t*-test. GraphPad Prism 8 was used for plotting. The results are represented as the mean ± standard deviation. * *p* < 0.05 indicated significant difference and ** *p* < 0.01 indicated extremely significant difference.

## 3. Results

### 3.1. Behavioral Differences in Female Rabbits at Different Stages of the Estrous Cycle

The estrus stage was determined according to the characteristics of the vaginal mucosa of the female rabbits. Daily behaviors of female rabbits at different estrous cycle stages were observed and recorded (Figure 2). As mentioned in Table 3, the duration of grooming behavior in estrus was significantly higher than in the diestrus (*p* < 0.01) and early estrus stages (*p* < 0.05). The duration of foraging and drinking behaviors were in contrast to that of grooming behavior, and the duration in diestrus was significantly higher than in estrus (*p* < 0.01, *p* < 0.05). There were no significant differences in the duration of relaxing, locomotion, standing, observing, and biting behaviors (*p* > 0.05).

The frequencies of behavior differences at different stages of the estrous cycle are mentioned in Table 4. The frequencies of relaxing and grooming behaviors in estrus were significantly higher than in diestrus (*p* < 0.05, *p* < 0.01). The frequency of drinking behavior was significantly higher in early estrus than in late estrus (*p* < 0.05). The frequency of locomotion behavior was the highest in estrus and significantly higher than in other stages *(p* < 0.05). The frequency of biting behavior increased first and then decreased with the estrous cycle, with the highest frequency observed in estrus.

### 3.2. Hormonal Differences in Female Rabbits at Different Stages of the Estrous Cycle

As shown in Table 5, the levels of FSH in the female rabbits first increased and then decreased during the estrous cycle. The FSH level was the highest in early estrus and significantly higher than in late estrus (*p* < 0.05). The differences in LH levels were similar to those of FSH. The LH level was the highest in estrus and significantly higher than in diestrus and late estrus (*p* < 0.05). The P_4_ levels in diestrus and late estrus were significantly higher than in estrus (*p* < 0.05). The E_2_ level was opposite to that of P_4_ and was significantly lower in early estrus than at the other stages (*p* < 0.05).

### 3.3. Follicle Development Differences in Female Rabbits at Different Stages of the Estrous Cycle

As shown in Figure 3A, the ovary structure in female rabbits consists of the white membrane (AI), cortex (Co), and medulla (Me). The Co and Me together form the parenchyma of the ovary. The cortex is located in the periphery of the ovary and contains follicles. The results revealed many lumen follicles (SFs and TFs) in estrus, but few in diestrus and early estrus (Figure 3B). There was no significant difference in the number of PrFs at all estrus stages (*p* > 0.05). The number of PFs in estrus was significantly higher than at other estrus stages (*p* < 0.05). The number of SFs in estrus was significantly higher than in diestrus and early estrus (*p* < 0.05). The number of TFs was higher in estrus and late estrus than in diestrus and early estrus (*p* < 0.05) (Figure 3C).

### 3.4. Screening of Ovarian Transcription Patterns in Female Rabbits in Diestrus and Estrus

To further understand the molecular regulation mechanisms of the estrous cycle in female rabbits, we analyzed the ovarian transcription patterns of female rabbits in diestrus (D group) and estrus (E group) using RNA-seq. A total of 967 DEGs were screened out between the D and E groups, among which 623 were up-regulated and 344 were down-regulated (Figure 4A,B). Several genes related to follicle development, gonadotropin synthesis, ovarian development, and ovarian steroid production were screened, including *BMPR1B*, *CYP19A1*, *PMFBP1*, *CYP1B1*, *OVGP1*, *IGF1R*, *EGFR*, *ESR1*, *ADCY7*, *ITPR2*, *SPP1*, *FGF23*, etc. Among these, *ESR1, IGF1R, CYP19A1, CYP1B1, BMPR1B*, and *HOXA9* were randomly selected to verify the accuracy of the transcriptome results via RT-qPCR, and the results were consistent with the sequencing results (Figure 4C). GO enrichment data analysis revealed that the DEGs were enriched in the positive regulation of estradiol secretion, reproduction, reproductive process, steroid hormone receptor activity, gonad development, etc. (Figure 4D). KEGG analysis revealed the DEGs mainly enriched in the estrogen signaling pathway, PI3K-Akt signaling pathway, MAPK signaling pathway, steroid hormone biosynthesis, and aldosterone synthesis and secretion pathways (Figure 4E).

## 4. Discussion

Animal behavior is the most direct manifestation of environmental change, and behavioral changes often reflect the internal physiological changes of the organism [22]. Animals modify their behavior in response to changes in their environment and physiological states, forming distinct time-stamped patterns and behavioral patterns [23]. The activity rhythm of the organism and behavioral time allocation can reflect individuals’ nutritional states and health status [24]. In this study, we observed that the duration of relaxing behavior decreased gradually from diestrus to estrus, but the frequency increased gradually. The foraging behavior of female rabbits was affected by estrus, and there was a significant decrease in the duration from diestrus to estrus. Some studies have pointed out a negative correlation between serum estrogen and feed intake [25,26], consistent with findings of increased estrogen concentration and decreased foraging time in estrus. Studies on weaned piglets also showed a significant positive correlation between foraging and drinking time [27]. When female animals entered the estrus stage, they became mentally sensitive and restless [28], and the frequency of grooming, locomotion, and biting behaviors increased significantly. Some studies have shown that animals behave more visibly after estrus [29]. It is assumed that animals have a short duration of estrus and need to mate in the shortest time, so they will show anxiety due to endocrine changes [30,31]. The daily behavior spectrum in female rabbits at different stages of the estrous cycle will provide more details for the identification of female rabbit estrus.

The growth and development of the follicles and the formation of the corpus luteum after ovulation are closely related to reproductive hormones [32]. The results of this study suggest that the role of FSH and LH may be related to how FSH and LH levels rise and subsequently fall during the estrous cycle, with LH peaking in estrus and falling in late estrus. Research has demonstrated that FSH and LH can synergistically stimulate granulosa cells to produce estrogen and facilitate follicle maturation [33]. The PFs, SFs, and TFs in estrus were significantly higher than those in diestrus. In addition, the TFs were significantly higher in late estrus than in diestrus and early estrus, and their peaks were consistent with the peaks of FSH and LH. The trend of E_2_ in the whole estrous cycle of female rabbits was the lowest in early and peaked in late estrus. This study showed the hormonal differences in female rabbits at different estrous cycle stages for estrus identification.

The estrous cycle of female animals is affected by external and internal physiological factors [34,35]. To explore the gene expression differences at different estrus stages in female rabbits, we analyzed the ovarian transcription patterns of female rabbits between diestrus and estrus. Several reproduction-related genes were found, such as *BMPR1B*, *CYP19A1*, *LHCGR*, etc. The overexpression of the *BMPR1A* and *BMPR1B* genes in ovarian granulosa cells inhibits the occurrence of mouse granulose tumors, while the knockout of *BMPR1A* revealed less spontaneous ovulation and reduced fertility in mice [36]. Studies have shown that *BMPR1B* regulates the differentiation of granulosa cells and follicle development by participating in the signal transduction of *BMP2*, *BMP4*, *BMP6*, *BMP7*, *BMP15*, and *GDF5*, thus affecting the reproductive cycle of mammals [37,38]. In particular, *CYP19A1* can convert androgens into estrogen, and estrogen can inhibit apoptosis in granulosa cells and promote its differentiation by increasing the expression of LHCGR, the LH receptor, thus regulating follicle development [39,40]. In this study, the expression level of *CYP19A1* was significantly higher during estrus compared to diestrus, with an increased change in plasma LH levels and a decrease in foraging and drinking times. These genes are potentially important in regulating the estrous cycle in female rabbits.

## 5. Conclusions

The female rabbits showed less foraging and drinking, and more grooming and biting in estrus. Related physiological hormones (FSH, LH, P_4,_ and E_2_) and follicular development with different estrous cycle stages also differed. Extensive information on behaviors, hormones, and follicle development in female rabbits at different estrus stages was investigated for appropriate postpartum re-estrus management strategies. The results provide recommendations for the adequate management practices of postpartum re-estrus and breeding female rabbits.

## Figures and Tables

**Figure 1 animals-13-03414-f001:**
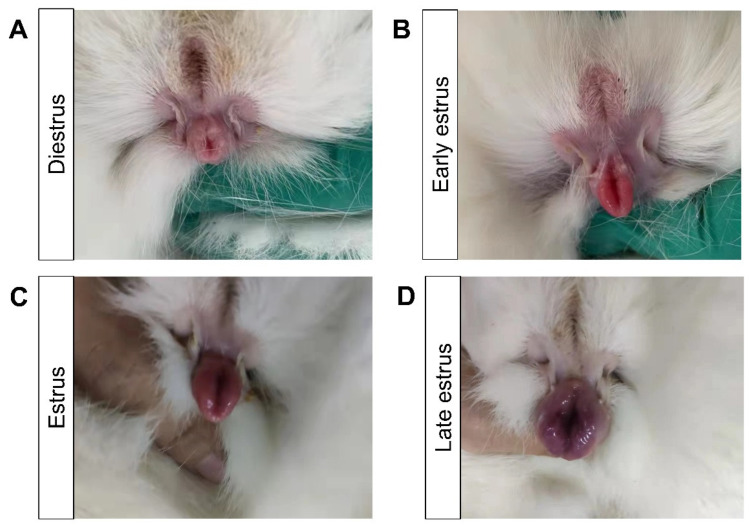
Vulva mucosa characteristics of female rabbits at different estrus stages: (**A**) diestrus stage; (**B**) early estrus stage; (**C**) estrus stage; (**D**) late estrus stage.

**Figure 2 animals-13-03414-f002:**
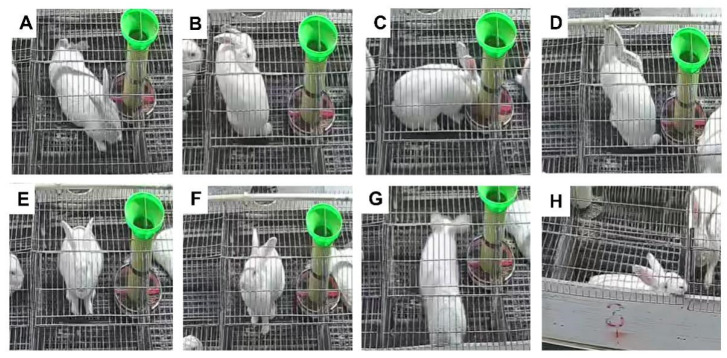
Video shots of female rabbits at different estrous cycle stages. (**A**) Relaxing behavior; (**B**) grooming behavior; (**C**) foraging behavior; (**D**) drinking behavior; (**E**) standing behavior; (**F**) observing behavior; (**G**), locomotion behavior; (**H**) biting behavior.

**Figure 3 animals-13-03414-f003:**
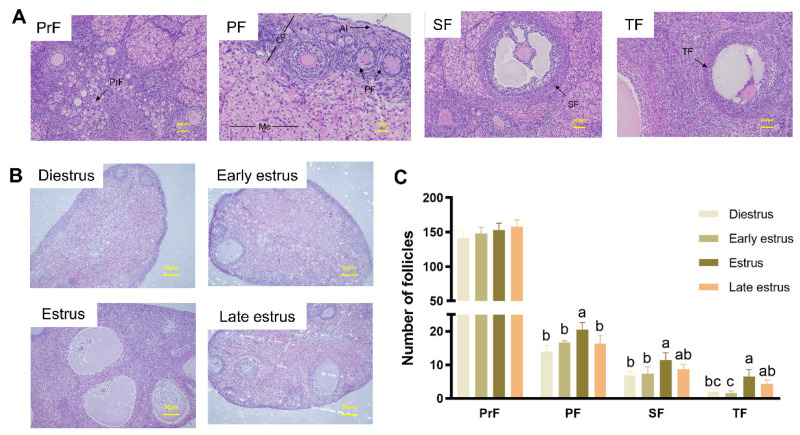
Ovarian histological observation of female rabbits at different estrous cycle stages. (**A**) Observation of follicle structure. PrF, primordial follicle; PF, primary follicle; SF, secondary follicle; TF, tertiary follicle; AI, albuginea; Co, cortex; Me, medulla. (**B**) Ovarian morphology of female rabbits at different stages of the estrous cycle. This included the diestrus stage, early estrus stage, estrus stage, and late estrus stage. (**C**) Follicle count of female rabbits at different stages of the estrous cycle. Different lower-case letters indicate significant differences between the groups (*p* < 0.05).

**Figure 4 animals-13-03414-f004:**
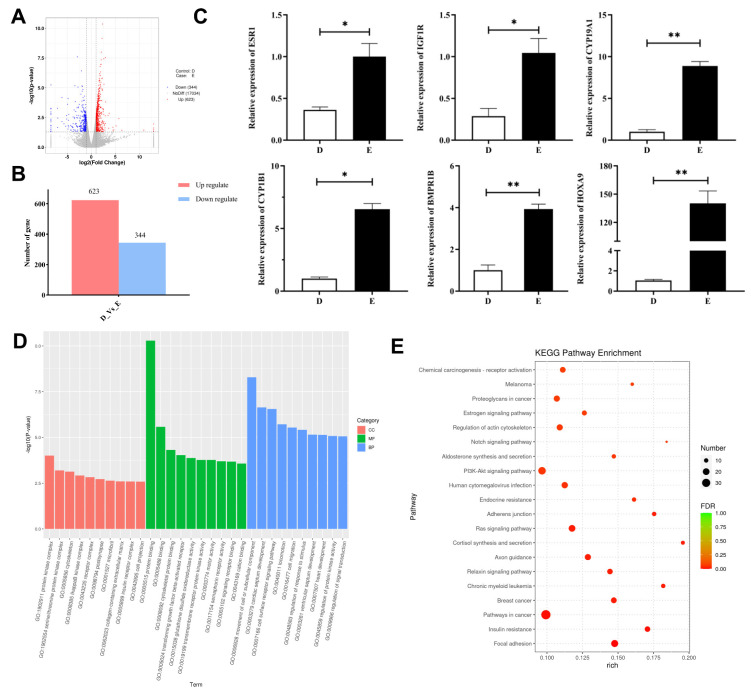
Screening of ovarian transcription patterns in female rabbits at diestrus and estrus stages. (**A**) The volcano map of the DEGs between diestrus and estrus. (**B**) The number of DEGs between diestrus and estrus. (**C**) GO classification map of the DEGs between diestrus and estrus. (**D**) KEGG signaling pathway classification diagram of the DEGs between diestrus and estrus. (**E**) Verification of mRNA expression levels of the DEGs between diestrus and estrus. * indicates significant difference (*p* < 0.05), ** indicates extremely significant difference (*p* < 0.01). D indicates the diestrus group, and E indicates the estrus group.

**Table 1 animals-13-03414-t001:** Daily behavior spectrum of female New Zealand White rabbits.

Behavior	Specific Description
Relaxing [16]	Lying on the bottom of the cage, limbs hidden under the body; or lying on the bottom, limbs spread out
Locomotion [17]	Moving the body, including running and jumping
Grooming [18]	Licking the fur and limbs and cleaning the face with the forelimbs
Foraging [16]	Bending the head close to the trough, and making a chewing motion after raising the head
Drinking [19]	Raising their head after approaching the water bottle and swallowing it
Standing [20]	Sitting and staying still without any purpose
Observing [16]	Standing on all fours, keeping the body still, and raising the head to look out of the cage or the next cage
Biting [21]	Gnawing hard objects, such as huts, sinks, and troughs

**Table 2 animals-13-03414-t002:** Primer information of the DEGs from RNA-seq.

Gene	Primer Sequence (5′ → 3′)	Accession Number	T_m_ (°C)	Product Size (bp)
*CYP1B1*	F: GCTGGGAACTGACTCCACTC	XM_002709757.4	60	143
	R: CAAGAACGCTTGGCTAGGGA			
*CYP19A1*	F: CCTGTCGTGGACTTGGTCAT	NM_001170921	60	279
	R: CACCTGGAATCGTCTCAGCA			
*BMPR1B*	F: GCAGTGGATGTGCCTTGTAT	XM_017347401.2	58	181
	R: CTCTTTCTTGGTGCCCACAT			
*HOXA9*	F: CCCCATCGACCCCAATAACC	NM_001171403.1	60	240
	R: GCCCGGTCCTTGTTGATCTT			
*IGF1R*	F: GAGGAAGCGGAGAGATGTCG	XM_051822976.1	60	251
	R: CAAAGTTGGAGGCACTGCAC			
*ESR1*	F: TGCTACGAAGTGGGAATGATGA	XM_051854226.1	60	273
	R: GGGTCTGGTAGGGTCGTACT			
*GAPDH*	F: CACCAGGGCTGCTTTTAACTCT	NM_001082253.1	60	145
	R: CTTCCCGTTCTCAGCCTTGACC			

**Table 3 animals-13-03414-t003:** Comparison of behavioral duration in female rabbits at different estrous cycle stages (n = 20, min).

Behavior	Estrus Stage			
	Diestrus	Early Estrus	Estrus	Late Estrus
Relaxing	845.50 ± 84.07	831.68 ± 130.06	819.90 ± 91.15	829.60 ± 119.96
Grooming	192.06 ± 47.91 ^Bb^	219.00 ± 54.65 ^ABb^	291.30 ± 13.60 ^Aa^	249.60 ± 39.65 ^ABab^
Foraging	171.42 ± 45.18 ^Aa^	133.06 ± 36.56 ^ABab^	98.86 ± 25.54 ^Bb^	105.40 ± 31.83 ^ABb^
Drinking	31.44 ± 13.64 ^Aa^	25.46 ± 8.61 ^ABab^	17.48 ± 6.81 ^ABb^	13.34 ± 5.74 ^Bb^
Locomotion	53.66 ± 38.27	47.10 ± 37.32	57.22 ± 27.03	49.74 ± 29.99
Standing	23.10 ± 14.56	11.18 ± 5.01	15.50 ± 6.88	19.26 ± 17.03
Observing	20.20 ± 17.37	22.80 ± 14.28	26.08 ± 10.42	20.86 ± 10.43
Biting	10.18 ± 3.32	8.86 ± 3.48	16.05 ± 12.07	9.17 ± 1.46

Different upper-case letters indicate a significant difference (*p* < 0.01), and different lower-case letters indicate a significant difference (*p* < 0.05). The same letter means no significant difference (*p* > 0.05).

**Table 4 animals-13-03414-t004:** Comparison of behavioral frequencies in female rabbits at different estrous cycle stages (n = 20, frequency).

Behavior	Estrus Stage			
	Diestrus	Early Estrus	Estrus	Late Estrus
Relaxing	99.00 ± 21.01 ^b^	103.40 ± 32.97 ^b^	139.80 ± 52.44 ^a^	114.80 ± 45.04 ^ab^
Grooming	97.80 ± 15.55 ^Bb^	124.00 ± 39.51 ^ABb^	177.60 ± 23.97 ^Aa^	132.60 ± 45.08 ^ABab^
Foraging	63.40 ± 12.50	56.80 ± 22.81	57.60 ± 7.57	55.20 ± 9.39
Drinking	40.40 ± 11.41 ^ab^	42.40 ± 10.92 ^a^	29.40 ± 11.55 ^ab^	26.80 ± 7.66 ^b^
Locomotion	37.00 ± 10.56 ^b^	44.50 ± 21.06 ^b^	74.75 ± 18.79 ^a^	36.75 ± 12.31 ^b^
Standing	28.25 ± 10.01	43.33 ± 19.22	28.75 ± 7.85	29.25 ± 11.53
Observing	19.75 ± 11.76	20.60 ± 7.09	30.80 ± 14.72	17.80 ± 6.72
Biting	10.00 ± 3.61 ^Bc^	18.33 ± 1.53 ^Bb^	34.00 ± 5.66 ^Aa^	23.50 ± 0.71 ^Ab^

Different upper-case letters indicate a significant difference (*p* < 0.01), and different lower-case letters indicate a significant difference (*p* < 0.05). The same letter means no significant difference (*p* > 0.05).

**Table 5 animals-13-03414-t005:** Reproductive hormone levels in female rabbits at different stages of the estrous cycle (n = 20).

Item	Estrus Stage			
	Diestrus	Early Estrus	Estrus	Late Estrus
FSH, mIU/mL	9.16 ± 0.52 ^ab^	9.48 ± 0.48 ^a^	9.06 ± 0.83 ^ab^	8.75 ± 0.52 ^b^
LH, ng/mL	50.37 ± 1.62 ^b^	52.33 ± 1.41 ^a^	52.83 ± 0.72 ^a^	48.97 ± 2.27 ^b^
P_4_, ng/mL	8.67 ± 0.51 ^a^	8.36 ± 0.53 ^ab^	7.94 ± 0.57 ^b^	8.75 ± 0.66 ^a^
E_2_, pg/mL	537.44 ± 36.65 ^ab^	524.99 ± 33.51 ^b^	554.07 ± 53.20 ^ab^	565.95 ± 46.14 ^a^

Different lower-case letters indicate a significant difference (*p* < 0.05). The same letter means no significant difference (*p* > 0.05).

## Data Availability

The datasets used and/or analyzed during the current study can be made available by the corresponding author on reasonable request.

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
