# Peer review of "Behavioral and Physiological Differences in Female Rabbits at Different Stages of the Estrous Cycle"

_animals, 2023, doi:10.3390/ani13213414_

Round 1

Reviewer 1 Report

Comments and Suggestions for Authors

Estrus involves complex physiological signs and behavior changes, but the exact ovulation time is difficult to determine which is most important to improve fertilities of rabbit. This manuscript was well designed and the results can support the conclusion. There are some issues should be addressed before accept for publication.

Major issues:

1.      As we know, rabbit belongs to induced ovulation animals. Four estrous stage was evaluated in this study. But the copulatory behavior is also an important stage and helpful to determine the ovulation time. So, I suggest this stage can be incorporated into experimental design.

2.      Transcriptomic sequencing is difficult to reveal the underlying mechanism of estrus behaviors. I know the authors want to investigate the transcription pattern related with follicular development. From the results we can found that these differential mRNAs are mainly involved in hormone synthesis etc. How to explain the behavior changes during estrus?

Minor issues:

1.      Line 80-82: The daily behaviors at different stages (diestrus, early estrus, prosperous estrus, and late estrus) of all female rabbits were monitored and recorded at 8:00 81am and 20:00 pm for 15 days. Why not cover the night time?

2.      From the figure 1, what can we know the duration of vulva mucosa color and each estrous stage?

3.      Please provide detail information of primers including Tm and product size etc. in table 2.

4.      I suggest table 3 and 4 can be replaced of figures and easy to see the behavior changes.

The quality of figure 3 and figure 4-D and 4-E is bad.

Reviewer 2 Report

Comments and Suggestions for Authors
Understanding the series of physiological and behavioral changes during the estrus process in rabbits is of significant importance for rabbit production and reproductive management. Currently, there is limited research available on the physiological and behavioral changes during rabbit estrus. The manuscript in question continuously observed the behaviors of 20 female rabbits over 15 days, investigating differences in feeding, grooming, resting, biting behaviors, as well as variations in reproductive hormone levels and follicle development at different stages of the estrus cycle in New Zealand does. Additionally, the study utilized transcriptome sequencing to analyze differences in ovarian transcript patterns between the proestrus and estrus phases in female rabbits. The research findings hold crucial reference value for the reproductive management of does in rabbit production. However, the analysis and discussion of the relationships between doe behavior, hormone levels, and gene expression are somewhat lacking in the article. It is recommended to make modifications and additions in this regard.
Other detailed suggestions for modification are as follows:
1. Line 52: The termits better  to replace "luteal body"  with "corpus luteum".
2. According to the description in the Materials and Methods section, "Four surveillance cameras were installed … each camera could record the activities of five female rabbits," and "The frequency and duration of each behavior … continuous recording method for 15 days". please confirm whether the behaviors of all 20 female rabbits were continuous recorded for 15 days. Analyzing continuous behavior for 15 days means  significant amount of work. Please provide details about the specific methods and software used to record and analyze rabbit behaviors.
3. Due to diurnal variations in hormone levels at different times of the day, please supplement the specific times of blood sample collection to enhance the reference value of the current experimental data.
4. It is recommended to supplement the durations of different stages of the estrous cycle observed in the experiment
5. The behaviors and their definitions in Table 1 for Grooming, Foraging, and Drinking are mismatched.
6. In the Discussion section, there is excessive repetition of the results, lacking analysis and discussion of the relationships among behavior, hormone levels, and gene expression. Suggest to modify and supplement this information.
7. In the References section, it is necessary to revise according to the uniform journal format and complete the citation information, such as References 16, 19 and 26. It is recommended to replace some old review literatures, such as References 15 and 28, with new progress researches.

Reviewer 3 Report

Comments and Suggestions for Authors

The abstract needs to be restructured. The authors mixed the methods with the results. Please change the sentences explaining the methods to after the sentence "The continuous recording method was used to record the daily behaviors in estrus cycle 12 of twenty postpartum female rabbits.". Then the results may be presented. The conclusion of the abstract should also be improved since it does not portrait a clear picture of what the authors achieved.

The introduction does a fairly good job of explaining the doe's estrus and some physiological changes, but little is written on the bahavioural aspects and studies. This should be expanded.

The materials and methods lack some important information. The authors have chosen to create their own ethogram and although they have taken care to pre-observe the animals in order to create these definitions, I believe there should be some literature sustaining their choices of behaviours. Also, the names should be equal to any available in literature not to produce more "clutter" and uniformize the information available in this line of research.

Results and discussion should be extensively reviewed. Some affirmations are a bit exagerated. Also, there is no conclusion for the manuscript.

L9: "However, there have been few studies that control for the stage of estrus in female rabbits.". The authors must mean that there are few studies that evaluate behaviour on each stage. The sentence needs to be corrected.

L40-45: These statements need to be corroborated by literature.

L66: The main aim of the manuscript needs to be clearer.

L77: individual cages? What sizes?

L101: Where the recordings observed by only one person? The frequency and duration of the behaviours was recorded manually or was any software used?

L104: Why not search for the behaviours that are already in literature? This should be justified in the text. I don't think the descriptons of each behaviour are correctly distributed.

L142-144: This informations should be in the materials and melhods section as well as figure 2.

L156: Was it measured in minutes? It should be explicit on the table.

L144-145: In fact this statement is not completely correct. Grooming and foraging were only significantly different between estrus and diestrus. Due to the high values for standart deviation and not so high number of animals, all of these behaviours end up being very close in duration time. I advise the authors to change the text to clarify specificaly which differences are statistical significant. Also, was data tested for normal distribution?

L171: Same analysis as before. The authors need to focus their text on objective statements on their findings. Most values end up being statistically equal.

L240: Based on this sentence I would suggest that the authors cautiously analyse correlations between the different factors analysed.

L244: "they became mentally sensitive and restless" based on what findings?

L246: Behave more..?

L248: Taking into consideration that most findings were statisticaly equal I advise the authors to change their approach and make statements that are much more cautious than this one.

Comments on the Quality of English Language

Moderate editing of English language required in some parts of the text.

Reviewer 4 Report

Comments and Suggestions for Authors

The study analyzes the behavioral and physiological difference of female rabbits at different stages of the estrous cycle. The paper is interesting and well written, despite it suffers for poor novelty in some aspects, but, overall, it adds new knowledge to the topic. Few, but important concerns must be added before it can be considered for publication.

The Simple Summary is missing.

The materials and methods must be improved by giving more details about the hormones assay, ELISA is too generic. The reader should be allowed to repeat the experiment by using the same reagents.

Statistics also need to be more detailed. The authors wrote “Statistical analyses were performed using the T-test, one-way analysis of variance….” It is not possible to understand the statistical approach. Please clarify the kind of test used for each analysis  by adding the relative formula.

A Conclusion section should be added to overall usefulness of the study in improving the knowledge on the topic and in opening new research perspectives.
